# Normal Percentiles for Respiratory Rate in Children—Reference Ranges Determined from an Optical Sensor

**DOI:** 10.3390/children7100160

**Published:** 2020-10-02

**Authors:** Anthony Herbert, John Pearn, Stephen Wilson

**Affiliations:** 1Children’s Health Queensland Hospital and Health Service, 4101 Brisbane, Australia; j.pearn@uq.edu.au; 2Children’s Health Queensland Clinical Unit, Faculty of Medicine, University of Queensland, 4101 Brisbane, Australia; 3School of Information Technology and Electrical Engineering, University of Queensland, 4072 Brisbane, Australia; wilson@itee.uq.edu.au

**Keywords:** respiratory rate, children

## Abstract

(1) Background: Increased respiratory rates (RRs) are described in several medical conditions, including pneumonia, bronchiolitis and asthma. There is variable methodology on how centiles for RR are derived in healthy children. Available age percentiles for RR have been generated using methods that have the potential themselves to alter the rate. (2) Methods: An optical respiratory sensor was used to measure RR. This technique enabled recording in awake children without the artefact of the observer’s presence on the subject’s RR. A cross-sectional sample of healthy children was obtained from maternity wards, childcare centres and schools in Brisbane, Queensland, Australia. (3) Results: RRs were observed in 560 awake and 103 sleeping children of which data from 320 awake and 94 sleeping children were used to develop centile charts for children from birth to 13 years of age. RR is higher when children are awake compared to asleep. There were significant differences between awake and sleeping RR in young children. The awake median RR was 59.3 at birth and 25.4 at 3 years of age. In comparison, the median sleeping RR was 41.4 at birth and 22.0 at 3 years. (4) Conclusions: The centile charts will assist in determining abnormal RRs in children and will contribute to further systematic reviews related to this important vital sign. This is particularly in relation to the data on children aged from 0 to 3 years, where data are presented on both the awake and sleeping state. Many studies in the literature fail to acknowledge the impact of sleep state in young children on RR.

## 1. Introduction

Counting and recording respiratory rate (RR) comprises one of the fundamentals of prehospital, medical and nursing assessments. RR is increased in both respiratory (e.g., pneumonia, bronchiolitis, and asthma) and nonrespiratory conditions (e.g., sepsis and ingestions) [1,2,3,4,5]. RR is important in monitoring children receiving opioid analgesics and an important early warning sign of the clinical deterioration of patients including children [6,7,8].

The medical literature and textbooks pay homage to RR as an important clinical sign. However, studies have found that RR is either not recorded by nursing staff or is poorly performed [9,10,11,12]. In addition, there is a variety of credible reference data on healthy ranges, especially in children. Many previously published ranges have methodological problems. This relates to issues associated with measurement methodology and locations which can modify the RR [13,14,15,16,17,18]. Key studies described in the literature are summarised in Table 1. A systematic review undertaken by Fleming et al. of previously published studies of RR has summarised and compiled many of these studies [19]. Of note, an Australian study measuring RR in healthy children and a study of the RR of children presented to an emergency department were not included in the systematic review undertaken by Fleming et al. [16,20].

More recently, large datasets have been extracted from electronic medical records. An Australian study analysed the heart rate (HR) and RR of 111,696 children presenting to a tertiary paediatric emergency department. These children were triaged as category 5 and were afebrile (temperature < 38 °C). This dataset did not include duplicates and the method of measuring RR was not stipulated. Bonafide et al. [21] also obtained 116,383 observations of RR and HR from 8894 patients from the electronic medical record of children admitted to two tertiary-care children’s hospitals. The state of the child (e.g., awake or asleep) and the methodology of measuring RR and HR were not stipulated, standardised or documented within the electronic medical record or study methodology. Percentile charts based on datasets derived from the Emergency Department (ED) or ward environment are invaluable, as clinicians need to assess infants and children in these environments. The setting of care for the patient (home, prehospital, ED, or inpatient) is therefore an important consideration.

The aim of this study was to produce percentile charts of RR for healthy children from birth to 13 years of age using a methodology which was accurate, noninvasive and applicable to common clinical scenarios. The use of a computerised respiratory sensor would minimise human and observer error. Such errors can relate to counting rapid or small chest wall movements, or respiratory movements masked by general body movements. RR is also sensitive to environmental conditions. States of arousal, temperature, sleep state, medications, monitoring equipment, and patient mental states will all affect RR [25,26]. Our secondary goal was to control for these various factors as much as possible. Further, we wanted to validate that counting the RR during two distinct 30 s intervals would give a similar result to the RR measured over a longer duration (e.g., 5 min). Finally, while Socioeconomic status (SES) is related to respiratory illness (e.g., asthma) and infection in children, there is no literature that suggests a relationship between SES and RR [27,28]. However, it was deemed helpful to determine how representative the current study sample was of the broader urban population within Brisbane.

## 2. Materials and Methods

### 2.1. Study Design

This is a cross-sectional study of children residing in the city of Brisbane, Queensland, Australia. Newborn babies were evaluated in maternity units of the Royal Women’s Hospital. Infants aged up to one year were recruited from a short term residential facility providing care for mothers and babies experiencing sleep or feeding difficulties. Children between the ages of six months and six years were tested in six different childcare centres located in four disparate geographical locations across Brisbane. Children aged five years and above were tested in four different schools and several community youth groups.

### 2.2. Ethical Approval

Ethical approval was granted by the ethics boards of the Royal Hospitals in Brisbane (children and women’s) (Protocol 94/21). Information on the child’s medical history, residential location and family socioeconomic status was collected from parental questionnaires.

### 2.3. Subjects

The study aimed to recruit 20 children per age group. Although, statistical advice was sought about this number before the study, there was little guidance available. This number was determined in terms of the practicality of completing the study over a 6 month period, and comparison with other similar studies of respiratory rates in children [16,20]. Age groups for awake children included each year up until the age of 12, and then 20 adolescents and young adults (AYAs). For sleeping children, the age groups included zero to one month, one to six month, six to twelve months, and then each year up until the age of 4 years. For this study, Brisbane was chosen for the study as it is represents a metropolitan region within Australia. Within Brisbane, participants were studied in 4 distinct geographical areas. These areas were also convenient locations for the investigator.

### 2.4. Experimental Protocol

The optical sensor used in this study detected respiratory movement through the angular movement of a hinge. It was developed for use in magnetic resonance imaging (MRI) and is described and validated elsewhere [29,30]. The sensor was ideal for a study of children as it is small, simple to use, easily attached, and caused minimal inconvenience or distress to the child. A diagram of the sensor is included in Figure 1. The sensor was attached to the lower costal margin and abdomen wall and detected the movement of the abdomen. Data points were sampled every 20 ms. Two 30 s recordings were used for the calculation of RR. The reason for using two thirty-second recordings of respiratory rate was because this is what will be most feasible in clinical practice [31,32].

All children had their axillary temperature, weight and height measured. The ambient temperature of the room was also measured. To maximise co-operation for the observation of respiratory rate, these measurements were taken after the children’s RR was measured. Axillary temperature was measured by thermometer (Safety First, Chestnut Hill, PA, USA). Weight was measured with scales (Salter, Kent, England). No further evaluation of the children was undertaken, so as to minimize any anxiety associated with the study [16].

Awake children were sedentary for 10 min prior to measurement. The sensor was attached at least five minutes prior to monitoring. The sensor was small and unobtrusive and hidden under the child’s clothing and attached with tape. The breathing pattern of each child was recorded for five minutes while the child was in a supine position. The children either read a set book themselves or were read to if under five years of age. Children were tested in a quiet room or corner of the childcare centre in the presence of a female assistant. Awake children were aware that their breathing was being measured. It was hoped that the story reading would distract the children from focusing on their breathing.

In sleeping children, the sensor was attached prior to sleep. Sleep state was categorised as quiet or active based on behavioural criteria [33,34]. Breathing pattern was recorded for five minutes during sleep categorised as quiet.

It was found that counting RR for two thirty-second periods gave a more accurate RR than RR counted for a single 60 s period [35]. In this context, two methods of measuring RR were recorded. RR was measured by counting the breaths detected by the device for 2 thirty-second periods. RR was also measured by counting the breaths over the total study period (approximately 5 min). The mean of the differences between measuring RR by these two methods was calculated to show if there was any systematic bias between the two methods. The standard deviations of the differences and the 95% limit of agreement between the two counting methods were also determined. Counting RR by adding respirations counted for 2 × 30 s periods is the method most likely to be used to determine RR in paediatric practice.

A questionnaire administered at the same time as the observations of respiratory rate excluded children with acute or chronic respiratory illness. Children with a history of pneumonia, croup or bronchiolitis were also excluded. This questionnaire also looked at the occupations of parents of participants in this study. Socioeconomic status (SES) was classified using the Australian Standard Classification of Occupations (ASCO) [36].

### 2.5. Statistical Analysis

Correlation coefficients (Pearson’s *r*) were used to examine the relationship between RR and age, weight, height, axillary and ambient temperature. Centile charts were constructed using a centile regression according to the method of Rashbak, Pan and Goldstein which has been endorsed by the World Health Organisation [37,38,39]. Grostat II, a computer program, was used to assist with this method [37]. This methodology was also used by Rusconi et al. to construct percentiles for RR in their study of children aged from birth to 3 years [18]. Logarithmic transformations were used to equalise standard deviations and variances across all age groups.

Data were displayed on a scatter plot according to ascending age. A “box” was placed on the left side of the scatter plot containing a predetermined number of data points. For awake children, this represented 34 data points, and for sleeping children 10 data points. A least-squares regression of RR on age was calculated for the data in the “box”. Centiles (3, 25, 50, 75 and 97) were then plotted for the “box”. The box then moved to the right by three data points and the process was repeated. Repetitions occurred until all data were analyzed. Data (half the size of the “box”) were lost for the youngest and oldest age groups.

The age range of interest was between one month and 12 years. A small number of children and young people outside of this age range were included to inform where the boxes would start and finish within the age range of study. As such, although smaller in number, the values of these children at the extremes of age assisted in guiding the centile curves in the areas of interest (0–3 years asleep and 1–13 years awake). Polynomial functions which reflected the raw data were used to smooth the curves.

## 3. Results

A total of 540 awake and 103 sleeping participants had their RR measured, of whom 197 awake children were excluded as they had a history of respiratory illness or cardiac disease, leaving data on 323 awake and 94 sleeping children, who were included in the calculations of reference ranges. This also included a further 20 adolescent and young adult (AYA) participants. Table 2 shows the age and gender of children measured. Figure 2 is a flow chart showing children who were excluded from the study and those included in the final evaluation.

It was observed that children with elevated respiratory rates had axillary temperatures greater than 37 °C, and seven children were therefore removed from the analysis. An infant on medication which could affect respiration and another child recovering from surgery were also excluded. Elevated temperatures did not affect RR in older children. Consequently, these children were not excluded from the study.

RR was reduced with age, with the greatest change between infancy and the age of three years. Table 3 displays two methods of counting respiratory rate. The mean of the differences between the RR determined using the two methods of counting RR was 0.279 bpm, indicating there was little difference (or bias). The largest reduction in RR occurs between age 0 (mean RR 41.4) and one year (mean RR 24.1)—see Table 4.

### 3.1. Centile Charts

A cubic equation was used to fit the raw centiles for RR in children who were sleeping (0 to 1 year), and children who were awake (1 to 3 years). A quadratic equation was used to fit the raw centiles in sleeping children (1 to 3 years), and awake children (3 to 13 years). Due to this methodology of creating the centile curves, the moving “box” of respiratory rates, the data at the upper ages are not represented (e.g., asleep children older than 3 years and awake children older than 13 years). The centile RRs calculated in awake children are summarised in Figure 3. Figure 4 summarises the RR centiles for children aged from birth to 3 years during quiet sleep. Details of cubic equations characterising the data are provided in Appendix A.

### 3.2. Representativeness of Population

A comparison of the occupations of parents within a small subgroup of children who participated in the study with those who did not was undertaken. Parents came from managerial and professional groups. Many parents (67%) who consented for their children to be in this study were from professional occupations compared with forty-four percent who did not consent (not significant, Chi-squares test).

Children tested with a past history of respiratory illness were excluded from the data used to create percentile curves for healthy children. This included 197 awake subjects and 27 sleeping children with a history of respiratory diseases such as pneumonia, asthma and bronchiolitis. Research on children, free from even minimal lung disease, is critical in the creation of reference ranges for healthy children [40].

## 4. Discussion

### 4.1. Comparison with Similar Studies

Most clinicians would agree that RR is a useful and important sign to measure [41]. There is significant variation between studies on RR in terms of whether the sleeping state is observed, the method of counting RR (e.g., visual observation, auscultation with a stethoscope or use of a monitor) and setting of study. RRs measured with stethoscopes or respiratory monitors are usually higher than those with observation alone [18,26]. This could be due to the counting of little breaths that may not be observed when looking at chest wall movements. Touching a child with a hand or stethoscope stimulates an increase in the RR [26]. Measuring RR can also be manual or automated. A noncontact method of determining RR in children has advantages in that it reduces the influence of external factors. The optical sensor used in this study was able to utilise such benefit. A systematic review of 20 studies by Fleming did not include all studies of children in their review, e.g., Marks et al. [16] and Hooker et al. [20] were not included in their review.

Despite having “Measurements taken at heights greater than 1000 m above sea level” as an exclusion criteria, a study performed above this altitude (Denver, CO, USA) by Iliff et al. was still included in the review by Fleming et al. [14,19]. The lower oxygen partial pressure at this altitude may have affected their respiratory rate. Further, in the study by Iliff et al., children aged less than 18 months of age were tested while they were sleeping. Between 18 months and three years of age, respirations were counted when children were either awake or asleep. Other studies accounted more specifically of whether the young child was awake or asleep [18].

Table 5 shows RRs obtained from the current study alongside RRs derived from Fleming et al. [19]. Median RRs are quite similar with the average of the difference in median RR for the different age groups being just 0.6. For the first centile, RR was higher in the current study up until age 4 where it then became lower. One explanation for this difference could be the presence of a lower RR for sleeping children within the data presented in the systematic review. Most observations of RR in sleeping children will occur under the age of 4 years [42]. For the 99th centile, except for ages 1.6 to 3 years, the RR from the current study was generally higher than in the study by Fleming et al. [19], with the average of the difference in RR for the 99th centile being −1.6.

Studies of the RR of children in ED and inpatients in wards have found higher RRs for children aged older than 3 years [21,24]. It is possible one reason for this is the anxiety associated with a visit to an ED, or alternatively the impact of illness on hospital inpatients. O’Leary et al. found lower respiratory rates in children aged less than 3 years compared to the study by Fleming et al. and Bonafide et al. [19,21,24]. Children aged less than 3 years of age are more likely to sleep during the day, and this is one important consideration when measuring RR in this age group [42]. This cannot fully explain the differences between various study results as it would be unlikely a young child would sleep during the triage assessment or presentation to an emergency department [24]. Nevertheless, wherever possible, it would seem important to note whether the RR is taken during sleep or when the child is awake.

A study of 293 awake and 123 asleep Australian children by Marks et al. was not included in the systematic review by Fleming et al. [19]. It is possible that this was due to the exclusion factor “Children with illnesses likely to affect the cardiac or respiratory system”. While Marks et al. did include children with past or current respiratory symptoms, they found this did not impact on RRs. This study is most similar to the current study in terms of design and results [19]. The current study and that by Marks et al. [16] were performed in Australia, with a single investigator and used a respiratory sensor and automated counting technique. Marks used a nasal thermistor which did not involve undressing the child, nor close physical contact with the investigator. However, it is possible a nasal thermistor may stimulate trigeminal receptors which are known to lower RR. However, limitations include not being able to record RR of mouth breathers and 50% of children aged between one and two years of age refusing to participate or becoming upset during the study. The RR for both studies was quite similar, although Marks had a larger age range for the asleep children—see Table 6.

### 4.2. Clinical Utility of Centiles for Respiratory Rate

Centile charts for RR should assist in differentiating between children with health and disease. Traditionally, centile charts are used to monitor growth in the same child over time. There may therefore be some limitations in using a centile chart to categorise a respiratory rate measured at one particular point in time [43]. However, it may be possible to observe a baseline respiratory rate in a child with a chronic disease (such as cystic fibrosis or a neuromuscular condition) and see how this changes over time. The reduction in baseline RR with growth and development may be less than that seen for healthy children.

It may also be possible to monitor changes in a child’s respiratory rate during the one acute episode of care. For example, an increase in respiratory rate may indicate worsening asthma, while a reduction may indicate response to treatment. Alternatively, a reduction in respiratory rate may indicate deterioration in a child’s condition (e.g., in a child with a brain injury or alternatively a child receiving excessive analgesia with an opioid infusion). RR can be combined with other observations (e.g., HR and blood pressure) to determine an overall assessment of the severity of a child’s acute illness. Such a composite score, or early warning score, can be a sensitive indicator, activating criteria and warning of early deterioration in an unwell child [44,45].

Based on the current study, and other recommendations in the literature, we suggest measuring RR for two separate 30 s intervals, as the preferred method of measuring RR [35]. Using pulse oximetry as another “vital” sign can result in important changes in the treatment of some children [46].

### 4.3. Importance of Consideration of Awake or Asleep State

The current study and the study by Marks et al. [16], contrary to the systematic review [19], suggest that sleeping RR in younger children may be a more specific and sensitive indicator of disease than RR in awake children. Figure 5 demonstrates the difference in RR for younger children during both awake and asleep states. Younger children do spend more time in the day sleeping, and there would be opportunity to observe sleeping RR in short stay wards or during overnight admissions [42].

Another area of interest relates to what centile to use for the upper limit of health, e.g., 97 or 99 centile. In the current study, we have used the 97th centile based on the precedent of growth charts [47]. We also feel it preferable to assign abnormality based on a lower centile or RR and rule out disease based on other physical signs and observations, investigations, and changes in the child’s condition over time.

While not having the power of larger studies derived from electronic medical records [21,24], or a systematic review, smaller studies such as the current study do contribute to the knowledge and understanding of RR in terms of their focus on the physiology and method of measurement of RR. The state of the child (awake and asleep) and method of measuring RR are important in this context. Further, for neonates and infants differentiating between active and quiet sleep may also be an important consideration [33,34].

### 4.4. Representative Study Populations

Studies of RR in healthy children within the community, alongside those of sick children or children within the hospital environment, will contribute to the overall knowledge and understanding of RR in both health and disease. It is important to seek the context, including both the study population and setting, when interpreting findings from studies of RR in children [43]. For example, RR might be raised by various factors such as a stressful clinical environment, fever and pain. It is also possible such effects may vary between ages. The study of reference ranges undertaken by Hooker et al. and O’Leary et al. in an emergency department and by Bonafide et al. in an inpatient setting has some appeal therefore in this context [20,21,24]. The current study had an increased proportion of children from professional and administrative groups. Recruiting children from childcare centres and private schools may have accounted for this. However, there was no statistically significant difference between those who participated in the study, and those who did not in a subgroup of the larger participant group.

### 4.5. Limitations

While the number of subjects in this study is similar to that of Marks et al. [16], more subjects in each age group would have been helpful. The number of children sleeping who had their RR measured was also small (averaging 10–20 children per age category). This is particularly relevant in the context of using a log transformation to obtain a normal distribution and using polynomial equations to smooth the centile charts. There are also alternative ways of determining centile charts. The current study used the method of Healy et al. [38]. Another method proposed by Cole et al. [48], also allows any centile to be quantified. Further, data at the extremes of ages (i.e., youngest and oldest children) may have more of a contribution to the centile chart [48].

Awake children were also informed about certain aspects relating to the study before their RR was counted. This may have caused the children to become aware of their breathing and therefore alter their breathing patterns. Another limitation of the study was the exclusion of patients with fever in the children who were asleep. Children with fever were not excluded from the awake group as their RR did not appear elevated. This has resulted in some inconsistency.

## 5. Conclusions

Having accurate percentile charts or reference ranges of respiratory rate for healthy children is important so that abnormal respiratory rates can be detected. A need for well-designed studies of respiratory rates has been highlighted [49]. This study provides reference ranges of RRs for healthy infants and children from birth to 13 years for awake children. These data are specific for Australian children and is particularly useful for sleeping infants less than 3 years of age. Some strengths of this study include the carefully designed methodology and close observation of whether children were asleep or awake. It may be helpful for institutions to suggest how RR is measured, and when documenting RR to note the state of the child (sleep or awake) for young children aged less than 3 years. This could further enhance the value of the excellent percentile charts obtained from large datasets from electronic medical records.

## Figures and Tables

**Figure 1 children-07-00160-f001:**
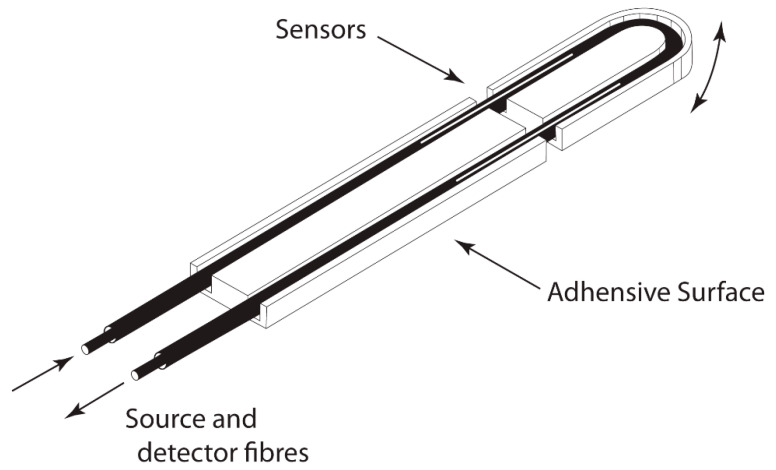
Illustration of optical respiratory sensor (the sensor is approximately 8 cm in length).

**Figure 2 children-07-00160-f002:**
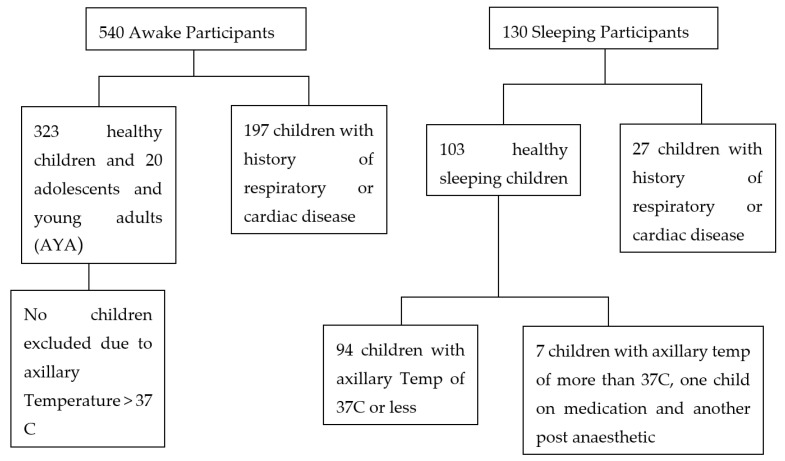
Flow chart showing children who were excluded from the stud and those included in the final evaluation.

**Figure 3 children-07-00160-f003:**
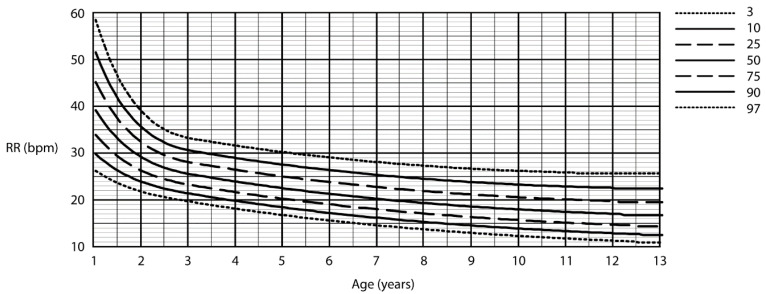
Respiratory rate centile chart in awake children (centiles—3rd, 10th, 25th, 50th, 75th, 90th, 97th).

**Figure 4 children-07-00160-f004:**
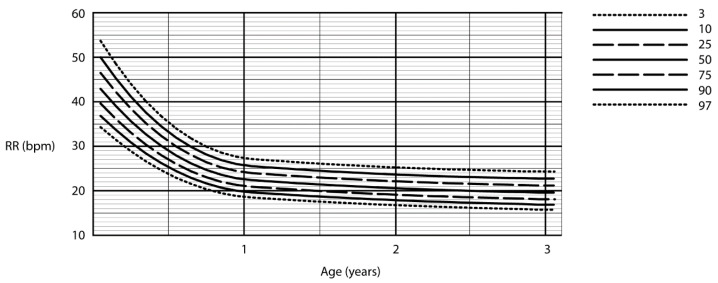
Respiratory rate centile chart in asleep children (centiles—3rd, 10th, 25th, 50th, 75th, 90th, 97th).

**Figure 5 children-07-00160-f005:**
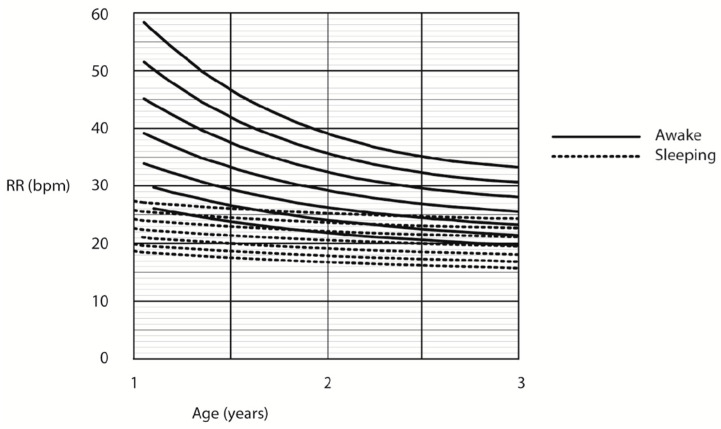
Respiratory rate centile chart in asleep and awake children (centiles—3rd, 10th, 25th, 50th, 75th, 90th, 97th).

**Table 1 children-07-00160-t001:** Previous studies citing reference ranges for respiratory rate (RR) in children.

Author	*N*	Age Range (Years)	Arousal State	Method of Measuring RR	Country
Quetelet (1842) [13]	300	0–20	Not known	Visual Observation	England
Iliff and Lee (1952) [14]	197	0–18	Awake	Observation	United States
Hooker et al. (1992) [20]	434	0–18	Awake	Observation in ED	United States
Marks et al. (1993) [16]	416	1–7	Awake/asleep	Thermistor	Australia
Rusconi et al. (1994) [18]	718	0–3	Awake/asleep	Observation (with stethoscope)	Italy
Wallis et al. (2005) [22]	1109	4–16	Awake	Observation	UK
Wallis et al. (2006) [23]	346	5–16	Awake	Observation	South Africa
Bonafide et al.(2013) [21]	116,383	0–18	Not specified	Observation	USA
O’Leary et al. (2015) [24]	111,696	0–16	Awake	Observation	Australia

RR—respiratory rate; ED—emergency department.

**Table 2 children-07-00160-t002:** Age and gender of healthy children in study.

Awake Children				Sleeping Children			
Age (Year)	Boys	Girls	Total	Age (Month, Year)	Boys	Girls	Total
0	12	6	18	0–<1 month	7	8	15
1	6	8	14	1–<6 month	7	12	19
2	11	8	19	6 month–<1 year	7	8	15
3	8	18	26	1	12	8	20
4	13	17	30	2	15	8	23
5	15	12	27	3–4	8	3	11
6	14	18	32				
7	13	21	34				
8	24	23	47				
9	13	7	20				
10	22	9	31				
11	5	9	14				
12	4	7	11				
13–19	5	2	7				
20–25	13	0	13				
Total	178	165	343		56	47	103

**Table 3 children-07-00160-t003:** Respiratory rate (RR) in awake healthy subjects calculated by two methods.

Age (Year)	*N*	RR (bpm) Measured for Two 30-s Periods					RR over Total Test Period		
		Mean	SD	Median	SE	CI (95%)	Mean	SD	Median
0.0–0.4	14	59.3	9.8	58.5	2.62	5.14	58.2	9.1	57.1
0.5–0.9	4	45.5	18.2	40.0	9.10	17.83	47.0	15.6	42.0
1	14	31.9	7.2	29.0	1.93	3.78	32.3	7.7	28.9
2	19	26.7	2.9	26.0	0.66	1.29	26.5	2.5	26.0
3	26	25.4	3.4	25.0	0.67	1.31	24.9	3.6	24.6
4	30	23.5	4.3	22.5	0.79	1.55	23.0	3.6	22.9
5	27	22.0	3.6	22.0	0.7	1.37	21.9	3.2	22.0
6	32	20.4	3.6	21.0	0.64	1.25	20.2	3.4	20.3
7	34	20.6	3.6	20.5	0.62	1.22	19.6	3.5	19.5
8	47	19.7	3.2	20.0	0.47	0.93	19.2	3.0	19.8
9	20	20.5	3.3	21.0	0.73	1.43	20.1	3.7	20.2
10	31	18.7	3.4	19.0	0.62	1.21	19.0	2.9	18.6
11	14	16.7	3.7	17.0	0.98	1.92	17.7	3.2	16.2
12	11	16.9	4.5	16.0	1.34	2.64	17.2	3.9	16.5

Bpm—breaths per minute, SD—standard deviation, SE—standard error, CI—confidence interval.

**Table 4 children-07-00160-t004:** Respiratory rate (RR) in healthy subjects during quiet sleep calculated by two methods.

Age (Year)	*N*	RR (bpm) Measured for Two 30-s Periods					RR over Total Test Period		
		Mean	SD	Median	SE	CI (95%)	Mean	SD	Median
0–1 week	10	41.4	4.1	41.0	1.29	2.53	41.3	3.7	41.2
1 week–1 month	4	41.5	5.4	40.5	2.72	5.34	41.1	5.5	41.2
1–6 month	17	35.4	7.2	34.0	1.74	3.41	34.2	7.9	32.0
6 month–1 year	13	24.1	2.8	23.5	0.75	1.47	23.9	3.0	23.8
1–2 year	17	22.1	3.5	21.0	0.85	3.49	21.9	3.4	21.2
2–3 year	22	19.5	2.7	19.0	0.57	1.11	19.5	2.6	19.3
3–4.5 year	11	19.3	2.7	18.5	0.85	1.65	19.3	2.8	18.3

Bpm—breaths per minute, SD—standard deviation, SE—standard error, CI—confidence interval.

**Table 5 children-07-00160-t005:** Comparison of awake RR data from current study and study by Fleming et al. [19] (1st Centile, Median and 99th Centile).

Age	RR 1st Fleming [19]	RR 1st Current Study	Δ	Median RR Fleming [19]	Med RR Current Study	Δ	RR 99 Fleming [19]	RR 99 Current Study	Δ
1–1.5	21	22	−1	35	35	0	53	55	−2
1.6–2	19	21	−2	31	30	1	46	44	2
2.1–3	18	19	−1	28	27	1	38	37	1
3.1–4	17	18	−1	25	25	0	33	35	−2
4–6	17	16	1	23	23	0	29	32	−3
6–8	16	14	2	21	20	1	27	30	−3
8–12	14	11	3	19	18	1	25	29	−4
12–15	12	NA	NA	18	NA	NA	23	NA	NA
Mean Δ			0.1			0.6			−1.6

Δ denotes the difference between RR in current study and RR in referenced study for a given centile. NA—not available.

**Table 6 children-07-00160-t006:** Comparison of RR data from current study and study by Marks et al. [16] (1st Centile, Median and 99th Centile).

Age	Expected RR (Mean) (Current Study)	3–97%	Expected RR (Mean) (Marks et al., 1993) [16]	2.5–97.5%
Awake				
1–2	31.9	23–46	31	22–43
2–3	26.7	20–35	27	20–38
3–4	25.4	18–32	25	18–34
4–5	23.5	15–31	23	17–32
5–6	22.0	13–30	22	16–30
6–7	20.4	10–28	21	15–29
Asleep				
1–2	22.1	17–26	21	16–27
2–3	19.5	16–25	20	16–26
3–4	---	---	20	15–26
4–5	---	---	19	15–26

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
