# Peer review of "Normal Percentiles for Respiratory Rate in Children—Reference Ranges Determined from an Optical Sensor"

_children, 2020, doi:10.3390/children7100160_

Round 1
Reviewer 1 Report
The authors clearly stated the rationale behind the importance of respiratory rate as a vital sign and the current evidence for the current reference ranges and their limitations.
The methodology chosen appears robust and reproducible.
The results were presented clearly though I would like to see a clearer representation (perhaps in flow chart form) of those children excluded from the study. It is not clear why the authors included the data for 20 participants older than 13 in their study (see table 2), especially the 13 boys/men over the age of 20! Whilst they clearly state that 343 children are included in the analysis, should the number not be 323.
The discussion is clear and to the point with appropriate limitations.
In the discussion of the clinical utility of centiles for RR the authors state that RR might be used during episodes of care though go on to contradict this, later in the same section, stating that "Caution must also be exercised in relation to directly imputing findings from studies of healthy children to the management of sick children"
Author Response
Please also see attached document which responds to all 3 reviewers.
Thank you for noting that our methodology appears robust and reproducible.
Children excluded from the study
- As suggested, we have given explanation of the children excluded from the study. Line 170-81.
- There were those with respiratory and cardiac illness. And then a group from the sleeping children group who had elevated temperature.
- A flow chart showing children excluded from the study has been included in the revised manuscript – see figure 2.
Comment on 20 participants aged older than 13 years
- In terms of the number of participants, the authors are correct, in that the number of children should be 323. There were 7 participants aged from 13 – 19 years and another 13 young adults included in the data. All of these participants gave informed consent for the study. The young men were medical students who volunteered during the early stages of implementation of the newly developed sensor within the clinical context (e.g. validation of the sensor – reported elsewhere, references 31, 32). The methodology used to create centiles uses a “box” method, and in this context, it was helpful to have a number of data points outside of the age range of focus 0- 13 years for the study. We have adjusted the number of patients presented in the study, and hope this clarification helps make the results more clear.
We have removed the comment "Caution must also be exercised in relation to directly imputing findings from studies of healthy children to the management of sick children"

Reviewer 2 Report
Overall
An interesting paper to read and will add to the literature
When I read the abstract about the ‘newly developed optical sensor’ I was expecting more of a validation study of the sensor, as well as the centiles. From reading the paper the sensor seems to have been validated already – see later note. I wonder if the whole emphasis of the paper should be on the derived centiles and deemphasise the sensor? Emphasising these, along with the variation between awake and asleep values makes this paper practically important when considering cut offs for scoring systems or activating criteria. I realise that only small numbers of sleeping children were captured and only to 3-4 years
Abstract
Results are a little confusing here. Quiet sleep at 1 month at 40, is compared to awake at 1 year ( 40), and quiet sleep at 3 years (20) to 13 years ( 12). It would be more useful to have the same age comparisons to show the difference, rather than to show the RR reduces with age – which is true but is well described already. Once again, hanging the emphasis to the difference between awake and asleep gives the paper more clinical impact. Consider also highlighting differences between studies
Conclusion: If the validation of the optical sensor wasn’t part of the study then it shouldn’t be part of the conclusion. Suggest you emphasise the main findings of this study as above. See comments on main work conclusion
Introduction
Good summary of the literature.
Methods
Authors should be congratulated for prospective data collection over multiple sites
Protocol line 92. If the sensor has been validated , ref 27 is from 1993, then it should be described as an already validated assessment method – are there any other references for this? This would be in line with comments above
The general methodology seems reasonable and effective
Statistical analysis
Lines 149-153 – I am not sure if some element of results has been included in the methodology. It appears that the inclusion of febrile or non febrile children was determined after the analysis based on whether the authors determined the fever influenced the RR. I am not sure of the validity of this. I would have thought methodology should have been determined prior to analysis and the safe approach would have been to exclude any child , regardless of age, from the analysis
It was observed that children with elevated respiratory rates had axillary temperatures greater 149 than 37 degrees C, and these children were therefore removed from analysis. An infant on 150 medication which could affect respiration and another child recovering from an anaesthetic were also 151 excluded. In older awake children, however, elevated temperatures did not affect the RR and these 152 children were not excluded from data analysis
The statistical methodology and analysis for centile derivation is complicated and I am not an expert in this. I have advised the editor that a statistical review should be sought, especially in relation to sample size and confidence intervals
I would like to see some confidence intervals around the centiles to see how confident these centiles are and whether there may be overlap between them, particularly when comparing awake and asleep.
Results
Paragraph Line 161 – two estimation methods – the methodology should be in methods, along with an explanation of why two methods were undertaken. I presume it is the show that in clinical practice a valid estimation is two 30 s periods?
Paragraph 189 – similar to above, methodology needs to be in method section. I am not sure if the cubic equation and quadratic equation are methodology or results – this may be my lack of statistical expertise with creating centile charts but the process needs to be clear to the average reader
Figures 2 and 3 represent the data well
3.2 – representativeness. I understand the need to describe the population, but is there a suggestion SE factors influence RR to make this relevant?
Discussion
Table 5 when viewed on own is difficult to interpret. Suggest clarification so that RR and RR current columns are obvious. RR current is the results from this study from the text
You discuss the results of Marks with awake and asleep. There would be value in comparing the results of this study asleep pts vs the Marks ones – similar to the way you have done with Fleming
It is hard for the readers to compare centile charts between papers, so if you do this it makes the paper more relevant and can add to your discussion
Additional visual comparisons that would add strength to the paper may be limited awake centiles – this paper, Fleming, Bonafide and O’Leary. This isn’t essential but I think would be really interesting
Conclusion
Suggest emphasise conclusions from this paper
Line 355 ‘in the context of a child’s overall condition’ – not sure how this is from this work
Lines 355 -360 really is a summary of the introduction
Suggest amplify the message from line 360 down, adding comparison and differences, need for asleep vs awake etc
This can then be part of the abstract, remembering that most readers will read the conclusion of the abstract and decide whether they want to even open the paper
Author Response
Please see attached document which summarises responses to all 3 reviewers.
Suggestion to focus on the respiratory rate centiles rather than the sensor
- We agree that the emphasis of the paper should be on the derived centiles, rather than validation of the sensor. In this context, we have referred to another reference where the sensor has been described and validated (eg references 31 and 32) . In this context we will also further emphasize the variation between awake and asleep values, and how this relates to cut offs for scoring systems or activating criteria. We also acknowledge the small number of children and smaller age range for sleeping children in our limitation.
Abstract
- We have emphasized the difference between RR in awake and asleep states at different age points (birth and 3 years) as suggested. Line 24 – 26.
- We have inserted the statement “There were significant differences between awake and sleeping RR in young children. Awake RR was 59.3 at birth and 25.4 at age 3 years. In comparison, sleeping RR was 41.4 at birth and 22.0 at age 3 years.”
- We also made a brief note on how different studies note (or fail to note) the difference between sleeping and aware RR especially in young children. [Lines 28-9].
- We have inserted the statement : Many studies in the literature fail to acknowledge the impact of sleep state in young children on RR. [Linse 37 – 38].
- We have removed the emphasis on the respiratory sensor in the conclusion of the abstract. [Removal of Lines 32-33].
Method
- Protocol – we have added in an additional reference which describes the validation of the sensor (e.g. references 31 and 32). [Line 102].
Statistical Analysis
- We have moved Line 149-53 to results. “It was observed that children with elevated respiratory rates had axillary temperatures greater than 37 degrees C, and these children were therefore removed from analysis. An infant on medication which could affect respiration and another child recovering from an anaesthetic were also excluded. In older awake children, however, elevated temperatures did not affect the RR and these children were not excluded from data analysis”. [Line 177-182].
- We agree that the febrile children should have been excluded from the study. However, we did determine the impact of fever on RR after analysis. We agree that this is a limitation, and have provided further explanation in the results, discussion and included a comment about this in the “Limitations” section. [Lines 177-182, Lines 383-5].
The statistical methodology and analysis for centile derivation is complicated
- We agree the methodology is complex. However, it is a method endorsed by the World Health Organisation (WHO) and described by Rashbask, Pan and Goldstein (1992) (Reference 39). They provided the computer program, Grostat II, was used to perform the calculations associated with this method (Ref 39). This method was also used by Rusconi et al (1994) to construct their centiles for RR in children aged from birth to 3 years (Ref 18). The method is outlined in more detail by Healy, Rashbash and Yang (Ref 40) and Pan, Goldstein and Yang (Ref 41).
- As the SD of RR can vary with age. A logarithmic transformation was used in an attempt to equalise SDs and variances across all ages group. Rusconi et al 1994, when creating percentiles for RR, also found this method to be helpful (Ref 41). [Line 154].
Confidence intervals around the centiles
- It would be difficult to place confidence intervals on the centile stage at this stage. However, we have provided the standard error and 95 % confidence interval in Table 3 and Table 4.
- We have also provided scatter plots of the raw data (figure 6.1 and 6.2). We have not included these in the manuscript in the interest of space.
Results
Two Estimation Methods
- Line 161 – two estimation methods – we have added this to the methodology section and an explanation of why two methods were undertaken i.e. to show that in clinical practice a valid estimation is two 30 s periods. [Line 132-140, Line 205-213].
I am not sure if the cubic equation and quadratic equation are methodology or results – this may be my lack of statistical expertise with creating centile charts but the process needs to be clear to the average reader
- We have added in information about the cubic and quadratic equation to the methodology section and results section, and we have tried to make this clearer to the reader. [Line 190-91] [Line 267-77].
Representativeness of the population
- As we were reporting on percentile charts for health children, we felt it important to make some comment on the representativeness of the population studied. While SE factors may influence RR illness, we are not aware of it impacting RR. Comments have been added to the introduction [Line 78-82], method [Line 163-5], results [Line 288-293] and discussion [line 465-469].
Discussion
- We have made Table 5 more clear.
- We have created a table comparing the results for sleeping children from this study with that of Marks (Ref 29). See Table 5.
- We agree with the reviewer that visual comparison would add strength to the paper, and would make the paper more relevant, and add interest. Due to the short time for review, we did not have time to do this.
Conclusion
- We have further reflected on the key conclusions of the paper (development of centiles for respiratory rate, consideration of obtaining a representative sample, and impact of sleep state versus awake state on RR). [Line 502-515].
Line 355 ‘in the context of a child’s overall condition’ – not sure how this is from this work
- We agree and have removed this statement
Lines 355 -360 really is a summary of the introduction
- We agree and have removed this
Suggest amplify the message from line 360 down, adding comparison and differences, need for asleep vs awake etc
- This is a good suggestion and we have done this [Line 502-515].
This can then be part of the abstract, remembering that most readers will read the conclusion of the abstract and decide whether they want to even open the paper
- We have also added this suggestion to the abstract [Line 30-37].

Reviewer 3 Report
- A brief summary(one short paragraph) outlining the aim of the paper and its main contributions.
The stated aim of this paper was to produce percentile charts of RR for healthy children from birth to 13 years of age using a non-invasive measurement method. The study provides valid reference ranges for RR for healthy children aged >1month to 3 years while asleep and 0-13 years while awake.
- Broad comments highlighting areas of strength and weakness. These comments should be specific enough for authors to be able to respond.
The manuscript is easy to read and has a good flow. The figures and tables are clear, easy to understand, stand alone and add value to the manuscript.
The discussion section “comparison with similar studies” lacks flow and seems repetitive. Consider rephrasing and consolidating it.
The non-contact method of determining RR in this population is definitely advantageous and valuable as it provides true RR reference ranges of young children that are largely unaltered by external factors.
- Specific commentsreferring to line numbers, tables or figures. Reviewers need not comment on formatting issues that do not obscure the meaning of the paper, as these will be addressed by editors.
Abstract & summary:
Line 16-17: “a newly developed sensor” but you go on to ref it in your materials & methods using a reference published in 1993. Please rephrase or discuss in your manuscript further
Line 24: I presume the values given here are medians? It would be useful to state it as you did in line 25
Introduction:
Line 43-44: “This includes a small number of subjects as well as issues associated with measurement methodology and locations which can modify the RR.” table 1 suggests that perhaps sample size (especially when compared to your sample size) is not truly an issue with previous studies, but perhaps measurement/method issues are. I would suggest rephrasing your sentence
Line 45-46: “Key studies described in the literature are summarised in Table 1. A systematic review of previously 45 published studies of RR is also included. [19].” – included in what? It is not included in Table 1 – please rephrase to improve clarity
Line 46-48: are you referring to the systematic review (ref 19) or your review (table 1) – this sentences clarity could be improved.
Line 62-69: Though your main aim is clear, I believe your manuscript would benefit from the addition of your objectives considering there are 3 main subsections in your results – i.e. comparison of RR calculation using 2 methods & evaluation of socioeconomic status
Materials & Methods:
Line 87-89: please provide a reference for these reported similar studies to support your planned sample size of 20 children per age group.
What are your age groups? Explicitly stating it would improve clarity
Line 126-128: why did you include socioeconomic status? Your introduction did not provide any background for this nor is it included in your aims. I recommend amending your intro & aims to provide a starting point for this inclusion – as it is currently, it seems out of place.
Results:
The total of sleeping healthy children in Table 2 is 103, but you state in line 157 that 94 sleeping children were included in the calculations for ref ranges. Why are these numbers not the same? Please amend this
Line 162-164: this information is better suited to the materials and methods rather than the results section
Line 165-167: please state that these results refer to RR in awake children
Table 3 & 4: were RR normally or non-normally distributed? Provision of this information would be useful to the reader to allow them to determine which central measure (mean or median) is more useful/accurate
Discussion:
Line 276-277 & 280-281: this is rather repetitious. Consider condensing this section
Line 285-286: “This could account for the lower RR observed in awake children at younger age ranges in the current study.” I assume you are referring to the Marks paper, but this sentence is not very clear. Consider rewording to prevent confusion about which study you are referring to; the current study (your study) or the study by Marks et al
Line 343-344: Yes, your group sizes are definitely a limitation and acknowledging more explicitly that especially the neonatal age group (while asleep) number is very low and likely reduces the validity of your findings/reference ranges for this age group. Further discussion about this is needed
Your study population includes 13 young adults (age 20-25yr) and 7 teenagers (13-19 yr) – the study’s stated aim is to develop RR centile charts for children from birth to 13 years of age (this is also reflected in the title). These 20 awake subjects should have been excluded from the study population. Please discuss why these subjects were included or exclude them from your analysis.
Given the importance placed on the sensor in the title and the study aim, some discussion of the optical sensor validity is warranted. As it been used to determine RR in children previously? How applicable do you think your centile charts/reference ranges are for use in clinical setting where people are likely to use contact methods (stethoscopes) or simple observation? Further consideration and discussion is warranted
Line 361-362: Though I agree with this conclusion, I do think it is important to exclude neonates from this statement as your sample sizes for this group was too small. The conclusion in its current form “This data is specific for Australian children and is particularly useful for sleeping infants less than 3 years of age” suggests that your results can be applied to any age of sleeping infants less than 3 years, but that is not the case. Please amend
Author Response
Please see the attachment which has responses to all 3 reviewers.
The discussion section “comparison with similar studies” lacks flow and seems repetitive. Consider rephrasing and consolidating it.
- We have done this to try and make this section more coherent.
The non-contact method of determining RR in this population is definitely advantageous and valuable as it provides true RR reference ranges of young children that are largely unaltered by external factors.
- We agree and have added this into the manuscript.[Line 257-60].
Line 16-17: “a newly developed sensor” but you go on to ref it in your materials & methods using a reference published in 1993. Please rephrase or discuss in your manuscript further.
We have removed the word “newly”
Line 24
- We have added in the word “median”
Introduction
Line 43-44: “This includes a small number of subjects as well as issues associated with measurement methodology and locations which can modify the RR.” table 1 suggests that perhaps sample size (especially when compared to your sample size) is not truly an issue with previous studies, but perhaps measurement/method issues are. I would suggest rephrasing your sentence
- Thank you for your suggestion, we have done this. [Line 42].
Line 45-46: “Key studies described in the literature are summarised in Table 1. A systematic review of previously 45 published studies of RR is also included. [19].” – included in what? It is not included in Table 1 – please rephrase to improve clarity
- We have re-phrased to improve clarity [Line 43-44].
Line 46-48: are you referring to the systematic review (ref 19) or your review (table 1) – this sentences clarity could be improved.
- We have improved this sentence [Line 45-47].
Line 62-69: Though your main aim is clear, I believe your manuscript would benefit from the addition of your objectives considering there are 3 main subsections in your results – i.e. comparison of RR calculation using 2 methods & evaluation of socioeconomic status
- We have added in comparison of RR calculation using 2 methods and consideration of socioeconomic status (SES) [Line 68-73].
Material and Methods
Line 87-89: please provide a reference for these reported similar studies to support your planned sample size of 20 children per age group.
We have provided 2 references – Marks et al and Hooker et al. (Ref 29,30) [Line 92-3]
What are your age groups? Explicitly stating it would improve clarity
We have included this. [Line 93-96].
Line 126-128: why did you include socioeconomic status? Your introduction did not provide any background for this nor is it included in your aims. I recommend amending your intro & aims to provide a starting point for this inclusion – as it is currently, it seems out of place.
We have included the following sentence at the end of the introduction.
- Further we wanted to compare to methods of counting RR and also evaluate socioeconomic status. While SES is related to respiratory illness (e.g. asthma) and infection in children, there is no literature that suggests are relationship between SES and RR[27,28]. It was felt helpful, however, to determine how representative the current study sample was of the broader urban population within Brisbane. [Line 70-73].
